# Surface Characterization and Tribological Performance Analysis of Electric Discharge Machined Duplex Stainless Steel

**DOI:** 10.3390/mi11100926

**Published:** 2020-10-07

**Authors:** Timur Rizovich Ablyaz, Evgeny Sergeevich Shlykov, Karim Ravilevich Muratov, Amit Mahajan, Gurpreet Singh, Sandeep Devgan, Sarabjeet Singh Sidhu

**Affiliations:** 1Mechanical Engineering Faculty, Perm National Research Polytechnic University, 614000 Perm, Russia; kruspert@mail.ru (E.S.S.); karimur_80@mail.ru (K.R.M.); 2Mechanical Engineering Department, Khalsa College of Engineering and Technology, Amritsar 143001, India; amitmahajan291@gmail.com (A.M.); devgan.sandeep186@gmail.com (S.D.); 3Mechanical Engineering Department, Beant College of Engineering and Technology, Gurdaspur 143521, India; singh.gurpreet191@gmail.com (G.S.); sarabjeetsidhu@yahoo.com (S.S.S.)

**Keywords:** material processing, DSS-2205 alloy, electric-discharge machining, surface integrity, wear resistance, surface wettability

## Abstract

The present article focused on the surface characterization of electric discharge machined duplex stainless steel (DSS-2205) alloy with three variants of electrode material (Graphite, Copper-Tungsten and Tungsten electrodes). Experimentation was executed as per Taguchi L18 orthogonal array to inspect the influence of electric discharge machining (EDM) parameters on the material removal rate and surface roughness. The results revealed that the discharge current (contribution: 45.10%), dielectric medium (contribution: 18.24%) majorly affects the material removal rate, whereas electrode material (contribution: 38.72%), pulse-on-time (contribution: 26.11%) were the significant parameters affecting the surface roughness. The machined surface at high spark energy in EDM oil portrayed porosity, oxides formation, and intermetallic compounds. Moreover, a pin-on-disc wear analysis was executed and the machined surface exhibits 70% superior wear resistance compared to the un-machined sample. The surface thus produced also exhibited improved surface wettability responses. The outcomes depict that EDMed DSS alloy can be considered in the different biomedical and industrial applications.

## 1. Introduction

Today, electric discharge machining notably established itself for the processing of hard and complicated geometrical contours, which are difficult to fabricate by traditional machining techniques [1,2]. This non-traditional machining technique showed its proficiency for the applications in the manufacturing of aerospace products, moulds, dies, etc [3,4]. The process is also recommended to fabricate the bio-implants owing to its favorable results in orthopedic fields [5,6,7]. The input process parameters namely pulse-on-time, pulse-off-time, current, dielectric medium, spark gap voltage, type of electrode and polarity (negative or positive) play a momentous role in the machining of diverse materials [8,9]. The optimum set of these machining performance parameters has promisingly enhanced the material removal rate and efficiently improves the surface properties [10,11].

Razavykia et al. [12] reported that the discharge current, pulse-on-time, electrode material, and voltage significantly influence the MRR and surface quality of Co-Cr-Mo alloy. Similarly, Mahajan and Sidhu [13] concluded that pulse-on-time, discharge current and electrode material were the dominant parameters for the improvement of corrosion resistance, wear characteristics and biocompatibility of Co-Cr samples. Furthermore, Philip et al. [14] employed the EDM for Ti6Al4V alloy and compared the tribological characteristics of a machined and unmachined specimen. The results of their study exhibited the improved specific wear rate and coefficient of friction due to the formation of oxides and carbide layers on the machined surface. Besides, Devgan and Sidhu [15] executed electro discharge treatment for investigating the surface wettability and corrosion resistance responses of β-titanium alloy. Simao et al. [16] scrutinized the impact of EDM process parameters on various responses such as material removal rate (MRR), tool wear rate (TWR) and surface hardness of AISI H13 tool steel. They concluded that the machining performance and surface properties of materials could be enhanced by an appropriate combination of EDM operation parameters. For instance, Singh et al. [17] reported dielectric medium and discharge current as eminent parameters for improved microhardness and wear resistance of stainless steel 316L using electro-discharge treatment. The processing of duplex stainless steel (DSS-2205) was carried out by Pramanic et al. [18]. They reported that input parameters i.e., pulse-on-time, pulse-off-time, and wire tension significantly influence the MRR, surface properties and kerf width. Alshemary et al. [19] reported that pulse-on and pulse-off time significantly influenced the wire-EDMing of DSS alloy. However, Rajmohan et al. [20] investigated the impact of wire EDM process parameters on the DSS-2205 alloy. It was observed that the current and pulse-on-time parameters had a considerable effect on MRR and SR. Rajaram et al. [21] utilized EDM for drilling of small holes (3 mm dia.) on DSS 2205. The experimental outcomes revealed that the input parameter such as current significantly contributed to the MRR of DSS alloy. Recently, Mahajan et al. [22] reported the excellent hemocompatibility and corrosion resistance outcomes of ED machined DSS-2205 alloy.

Along with the conventional tool electrodes, researchers also explored the performance of composite electrodes for the machining of hard to machine materials. Khanra et al. [23] considered a ZrB_2_-Cu composite electrode for machining of mild steel workpiece and reported the improved MRR and diminished TWR as compared to the Cu tool. Similar findings were observed by Tsai et al. [24] who utilized Cr/Cu composite electrode and confirmed the formation of a recast layer on the surface that improved the corrosion resistance. The results also demonstrated higher MRR and lower TWR as compared to other metal electrodes. Grisharin et al. [25] observed the improved wear resistance and machining efficiency when a copper-colloidal graphite composite electrode was used to machine different alloys. However, Teng et al. [26] employed a Cu-Ni composite tool for the processing of polycrystalline diamond specimens and suggested better MRR as well as surface roughness responses as compared to the Cu electrode.

According to the literature survey, as briefly discussed above, it was observed that various materials had been machined using EDM. However, this technique is not considerably reported yet for the machining of duplex stainless steel (DSS-2205) alloy. DSS-2205 can be used as an alternative for austenitic stainless steel (316L) owing to its enormous applications in industry as well as in the biomedical field. This paper reported the effect of different types of tool materials and dielectric medium on material removal rate, surface roughness, morphology, phase transformation, tribological performance, and surface wettability of the EDMed surface. The first step examined the effect of chosen process parameters on the MRR and SR of machined samples, and statically scrutinizes the significant factors. The next step studied the surface morphology and phase analysis of samples depicting superior results using field emission scanning electron microscopy (FE-SEM), x-ray diffractometer (XRD) and energy dispersive x-ray analysis (EDX) techniques. The DSS alloy is commonly utilized in mining industries, heat exchangers and oil or gas processing industries, where wear characteristics and surface wettability play an important role for long term usage of alloy. Therefore, contact angle measurement and pin-on-disc wear tests were performed and compared with the results with an un-machined sample to investigate the tribological and wettability behavior of the EDMed sample.

## 2. Material and Methods

### 2.1. Tool and Workpiece

In this research, duplex stainless steel (DSS-2205) in the form of a square plate of 90 mm with a thickness of 20 mm was procured from Solitaire Impex, Mumbai, India. The workpiece chemical composition of Fe: 69.93%; Cr: 22.81%; Ni: 5.2%; Mo: 3.05%; Mn: 1.43%; Si: 0.5%; C: 0.028%; P: 0.03%; S: 0.02%, and density 7.8 g/cm^3^; melting point 1350 °C, thermal conductivity 19.4 at 100 °C·W/mK, and electrical resistivity 0.085 × 10^−6^ Ω cm. Duplex stainless steel consists of chromium as its main content after iron and molybdenum which make greater utility of DSS-2205 alloy in the biomedical domain. The traditional machining processes are inappropriate to handle such hard materials. Therefore, under such circumstances, spark erosion commonly known as EDM employed as an emerging technique for treating such hard materials [27,28].

In this study, three different tool electrodes viz. graphite (C), copper-tungsten (25-Cu/75-W) and tungsten (W) were chosen for treating the DSS substrates in die-sinking EDM. Table 1 listed the specifications of all three electrodes. Initially, the emery paper (material: silicon carbide (SiC), grit-800) was employed for the surface finishing of the alloy plate. Further, the plate surface was cleaned with ethanol solution (C_2_H_5_OH) before ED machining.

### 2.2. Design of Experiment

The Taguchi methodology was used to design the experimental array. In this investigation, an orthogonal array of L18 mixed-level design matrix was used to scrutinize the effects of five controllable parameters on two responses i.e., MRR, and SR. The chosen process parameters and their corresponding levels are tabulated in Table 2. The Minitab-17 statistical software was used to prepare the experimental design matrix. Further, analysis of variance (ANOVA) was utilized to analyze the dominance of process parameters on the MRR and SR.

### 2.3. Experimental Procedure

All the experimental trials were performed on a die-sinker EDM (Electronica, India: Smart ZNC S50) with constant gap voltage (140 V) and machining depth (0.5 mm) for each run. Also, negative polarity (tool (+) workpiece (−)) has opted throughout the experimentation. The material removal process during the EDM technique depends upon the generation of heat on the substrate due to the abundance of electric sparks between the electrode [29,30]. Both tool electrode as well as alloy substrate immersed in the dielectric fluid tank that provides proper stability during this thermo-electric process.

An in-house fabricated tank (18” × 18” × 24”) was used of capacity 10 liters, containing a stirrer and circulation pump for appropriate flushing and avoiding debris within the working area. Figure 1 represented the schematic arrangement of EDM, experimental set up of machining and FE-SEM image of un-machined DSS-2205 substrate.

### 2.4. Calculations of Material Removal Rate (MRR) and Surface Roughness (SR)

The weight of the workpiece was measured before and after each trial using a precise weighing balance (Citizen CY220) for calculating the MRR using Equation (1).
(1)$$MRR({\mathrm{mm}}^{3}/\min.)=\frac{(w_{2}-w_{1})\times 1000}{\rho \times t}$$
where;
“w_2_ and w_1_” correspond to the workpiece weight (g) before and after each trial,“*ρ*” is the density of workpiece, and“*t*” (minutes) is the machining time.


The other output response i.e., surface roughness of the machined DSS substrates were measured using the Mitutoyo SJ-201 surface profilometer. The roughness of each machined sample was measured diametrically at three different points, and an average value was considered (Ra) for further investigation.

### 2.5. FE-SEM, EDX and XRD Analysis

After machining, the morphology of the surface was examined using FE-SEM (Hitachi SU-8810, Japan) at 11.0 kV of accelerating voltage. FE-SEM was also utilized to examine the surface after the wear test and also for the recast layer thickness. The phase transformation analysis (XRD; PANalytical X’Pert Pro MPD, The Netherlands) was performed using Cu-Kα X-ray radiation, and with generator settings of 40 mA and 45 kV. The elemental composition of the machined DSS-2205 specimen was analyzed via energy-dispersive X-ray spectroscopy (EDX; incorporated with FE-SEM) to observe the EDMed samples.

### 2.6. Investigation of Tribological Characteristics

Additionally, the sample exhibiting superior output responses i.e., high material removal and roughness were investigated for their tribological performance using a pin-on-disc type tribometer (DUCOM Instrument, Bangalore, India). ASTM G99-17, a standard for pin-on-disc wear analysis was followed and the EN31 steel disc of diameter 120 mm and thickness 20 mm was used to test the wear of the specimens at 100 rpm rotation speed. Test lubricant (ringer solution), track diameter (80 mm), steady load (70 N) and running time (3600 s) remains constant for each experimental run. The working operation of the tribometer, and calculations of wear, friction values obtained via associated software (TR-20LE) built-in with the attached computer system.

### 2.7. Microhardness and Recast Layer Thickness Measurement

German made; Mitutoyo microhardness tester was used under low-force hardness scale (HV 0.2) with a test force-load of 1.96 N for a dwell time of 10 s. The microhardness was figured thrice at distinct points, and an average value was noted for the calculation. The EDMed sample with superior output responses and correlated to tribological performance was cut cross-sectionally for measuring the recast layer thickness. The diamond paste was utilized for the mirror-polished of the substrate. The surface morphological investigation of a cross-section of the EDMed substrate depicted the recast layer thickness that was measured at five different positions at the transverse section and its average value was recorded.

### 2.8. Surface Wettability (Contact Angle Measurement)

The surface wettability of the alloy is the crucial property that impacts the other significant characteristics and also influences the enduring usage of the alloy substrate [31]. The hydrophobic or hydrophilic nature of the surface represented the wettability which was measured by the water contact angle (WCA). If WCA is greater than 90°, the surface is represented as hydrophobic, whereas, the angle lesser than 90° with the surface is considered as hydrophilic [32]. The wettability investigation was executed by utilizing a contact angle goniometer (Model 790; make: Rame–Hart instrument, USA) where the contact angle was computed in an environmental chamber through the sessile drop technique at 28 °C. The surface was cleaned with acetone solution before the experimentation. The contact angle of the substrate was measured at five different positions and an average value considered and reported as WCA. The digital camera captured a 20 μL distilled water profile of the droplet set on top of the substrate surface by a Gilmont microsyringe.

## 3. Results and Discussion

The present study predicts and optimizes the ED machining performance parameters for duplex stainless steel (DSS-2205). All the experimental trials were carried out thrice (i.e., 18 × 3 = 54 runs) to minimize the error and for precise outcomes. The respective results in Table 3 signified the average material removal rate and surface roughness values attained from each experimental run, followed by the standard deviation (i.e., Avg. ± S.D). Further, these results were investigated statistically via ANOVA and the most significant parameters that influence these outcomes were investigated.

### 3.1. MRR Results Investigation by ANOVA

Table 4 detailed the ANOVA results for the material removal rate of DSS-2205 alloy. The F-values, with a confidence level of 95%, acquainted the influential factors that extremely affect the substrate surface responses after machining. The parameters with higher F-value reveal its superior impact on the output machining responses. Likewise, the *p*-value indicated the significance level of the input controllable factor. The signal-to-noise ratios (S/N ratios) results of MRR represented current as of the most significant factor that majorly contributes in removing the material from DSS-2205 alloy, followed by the dielectric medium and electrode material.

However, the *p*-values (>0.05) for pulse-on-time and pulse-off-time are not considerable; consequently, both are in-significant factors for machining of DSS-2205 under the selected range of parametric settings. The machined alloy sample as per the parametric settings of trial 9, demonstrates the highest material removal rate (39.4 ± 0.98 mm^3^/min). These results are in accordance with previously reported studies, where researchers reported discharge current, pulse-on duration, dielectric type and electrode material as the most significant factors for the EDM performance affecting the output responses [33,34,35]. The optimum parameters for maximum material removal rate of DSS substrates were examined from the S/N ratios plot (Figure 2) as 16 A of current, P-on = 150 µs, P-off = 60 µs, and use of W-Cu electrode in EDM oil.

### 3.2. SR Results Investigation by ANOVA

Table 5 depicts the analysis of variance results for the input factors in order to observe their dominance affecting the S/N ratios outcome of the surface roughness. From ANOVA results, electrode material (*p*-value: 0.001) was the most significant factor with a confidence level of 95% that influences the surface roughness of the machined substrates. The other factors such as, pulse-on-time (*p*-value: 0.003), current (*p*-value: 0.011) and dielectric medium (*p*-value: 0.033) also play a momentous role in producing the rough surfaces. The S/N ratios plot (Figure 3) disclosed that the DSS alloy samples machined in EDM oil with copper- tungsten (Cu/W) electrode at 16 A current, 150 µs pulse-on-time with a 60 µs pulse-off-time provided the more substantial surface roughness responses.

Among all the trials, specimen machined according to the parameter set of trial 8 exhibits a highly rough surface (Ra = 1.4 ± 0.08 µm). The results showed that some other machined substrates also have improved surface roughness as compared to the un-machined surface (Ra = 0.64 µs). These outcomes also endorsed the prominence of EDM in the biomedical field, where surface roughness plays a crucial role in the adequate engagement of human tissues and bones with implant surface [36,37,38].

### 3.3. Surface Morphology and Compositional Analysis of Machined Surface

Figure 4 illustrates the surface morphology of the EDMed substrates (trial 8 and trial 9) exhibiting excellent results of material removal rate and surface roughness. Both the machined substrates showed micro and macropores and re-solidified metallic droplets on the surface. It is observable from the images (Figure 4a,b) that higher spark energy (Spark Energy = Current × P-on × Voltage) generate pores along with small peaks and valleys on the EDT surface [39,40]. The presence of the pores and molten metal droplets on the surface promotes the biological performance of the substrates [41].

The EDX spectrum of the EDMed sample (trial 9) exhibiting higher material removal and surface roughness represented in Figure 5a. The presence of basic elements of DSS alloys viz. Fe, Ni, Cr, Mo, Mn and C was observed on the surface. Moreover, apart from base elements of alloy, the high percentage of oxygen element was also observed on the machined substrate. The EDX outcome of phase transformation of the machined substrate is also affirmed by the XRD pattern (Figure 5b). The formation of compounds viz. rhombohedral structured iron oxide (Fe_2_O_3_), major phase of CrMn_1.5_O_4_, tetragonal structured iron-chromium (Fe-Cr), hexagonal structured chromium oxide (Cr_2_O_3_) and tungsten carbide (WC) on the machined surface improves the wear resistance of the alloy. The presence of these oxides and carbides on the surface also improves the biocompatibility of alloy substrate due to which DSS alloy can also utilize in the biomedical domain.

### 3.4. Tribological Performance Analysis of Machined Surface

The machined sample with superior outcomes i.e., trial 9 was further assessed for their tribological performance. Moreover, the wear rate and coefficient of friction of the EDMed substrates were compared with the untreated substrate of DSS alloy. Table 6 demonstrates the wear characteristics of both treated and untreated substrates. It has been noticed that the wear rate of an untreated substrate (3.52 ± 0.15 × 10^−5^ mm^3^/Nm) was higher than the EDMed substrate (1.23 ± 0.11 × 10^−5^ mm^3^/Nm). Figure 6a represented the wear rate comparison of both specimens, whereas, the coefficient of friction with time for both pin substrates is showed in Figure 6b. It has been portrayed that the co-efficient of friction (µ) value of the un-machined specimen (µ_average_ = 0.32) is greater than the EDMed substrate (µ_average_ = 0.23).

Wear appearances for the EDMed and un-machined specimens were depicted by FE-SEM images. The un-machined surface was witnessed with flakes, pits and deep grooves (Figure 7a). However, the machined specimen was found with black patches that symbolized the tribochemical reaction (specimen surface wear in the high oxide atmosphere) (Figure 7b). Moreover, the machined surface was noticed with light scratches and no delamination that reveal the high wear resistance of the substrate [42,43,44]. Evidently, electric discharge machining at elevated temperature results in the chemical reaction between the dielectric fluid and the workpiece material elements. It results in the formation of carbide and oxide layers on the substrate that improves the wear resistance of the surface.

### 3.5. Analysis of Microhardness and Recast Layer Thickness

Micro-hardness results are also in-line with wear resistance outcomes. EDMed DSS substrate (trial 9) showed 487 HV_0.2_ micro-hardness, which was 1.46 (approx.) times better than the un-machined surface (334 HV_0.2_) of DSS alloy. These results were further affirmed by the investigation of the recast layer thickness of the machined surface. The cross-section image of the treated substrate was witnessed with the thick recast layer (53.952 µm) on the surface (Figure 6c). The re-deposition of melting material droplets from a working specimen as well as a tool electrode could be the possible reason for the recast layer on the machined substrate [45,46]. The outcomes portrayed improved resistance towards wear and micro-hardness of the machined substrate in comparison with an untreated substrate that admires the EDMed substrate in various industrial applications.

### 3.6. Surface Wettability Analysis

The WCA scrutinization demonstrated the hydrophobic or hydrophilic nature of tested substrates. The contact angles procured on untreated and treated substrates after 10s are represented in Figure 8. The results clearly showed that the EDM considerably improved the wettability of the surface. The WCA of the machined surface was 78.27 ± 0.41° respectively signifying the hydrophilic surface. The un-machined surface was hydrophobic with WCA of 105.96 ± 0.52° (above 90°). These results were in accordance with surface roughness outcomes. However, the surface roughness of the substrate has a huge impact on the wettability of the surface. Some researchers reported the direct relation between surface roughness and wettability [47]. The increased surface roughness leads to enhance the surface free energy by offering the expanded surface area to water droplet [48,49]. Therefore, surface alteration enhances the wetting responses of surface that endorse the machined substrate applications in the biomedical domain, lubrication and for different coatings [50,51].

## 4. Conclusions

The present study described the processing of duplex stainless steel (DSS-2205) alloy by EDM using graphite, copper-tungsten and tungsten as electrodes in two different dielectric mediums, namely EDM oil and deionized water. Based on the result, following conclusions have been drawn.

The dominating factors depicting maximum material removal rate (39.4 ± 0.98 mm^3^/min) were current (contribution: 45.10%), dielectric medium (contribution: 18.24%), and electrode (contribution: 12.67%).

The higher surface roughness accelerates the osseointegration process of bioimplant. The most significant parameters for higher surface roughness (Ra = 1.4 ± 0.08 µm) were electrode (contribution: 38.72%), pulse-on-time (contribution: 26.11%), and current (contribution: 17.39%).

From the S/N ratios plot, the optimum parametric combinations for favorable MRR and SR values by ED machining of DSS substrate are, EDM oil as dielectric medium, W/Cu electrode and current at 16 A, pulse-on-time at 150 µs coupled with the lowest pulse-off-time (i.e., 60 µs).

The FE-SEM and XRD examination confirmed the evenly distributed porous surface, and formation of oxides and other intermetallic compounds on the DSS-2205 surface machined at higher spark energy in the presence of EDM oil as a dielectric medium.

Moreover, EDMed substrate with higher MRR and surface roughness also exhibited enhanced wear resistance and surface wettability responses as compared to untreated DSS alloy substrate. The results also illustrated that EDMed DSS-2205 could be employed in the biomedical field.

## Figures and Tables

**Figure 1 micromachines-11-00926-f001:**
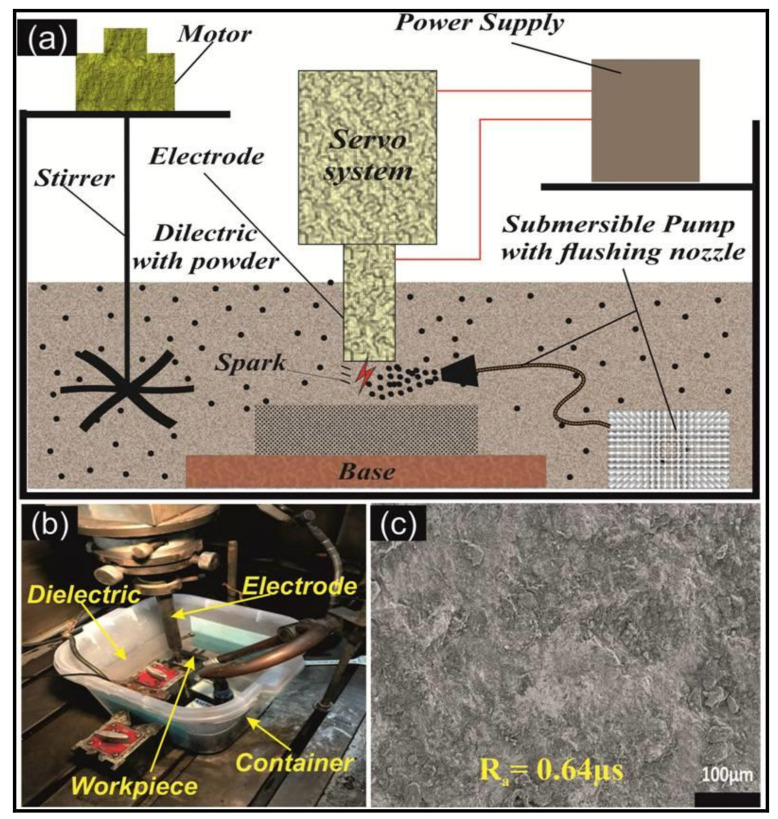
(**a**) Schematic arrangement of EDM; (**b**) Pictorial view of experimental set up of machining; (**c**) FE-SEM image of un-machined DSS-2205 substrate (Ra = 0.64 µs).

**Figure 2 micromachines-11-00926-f002:**
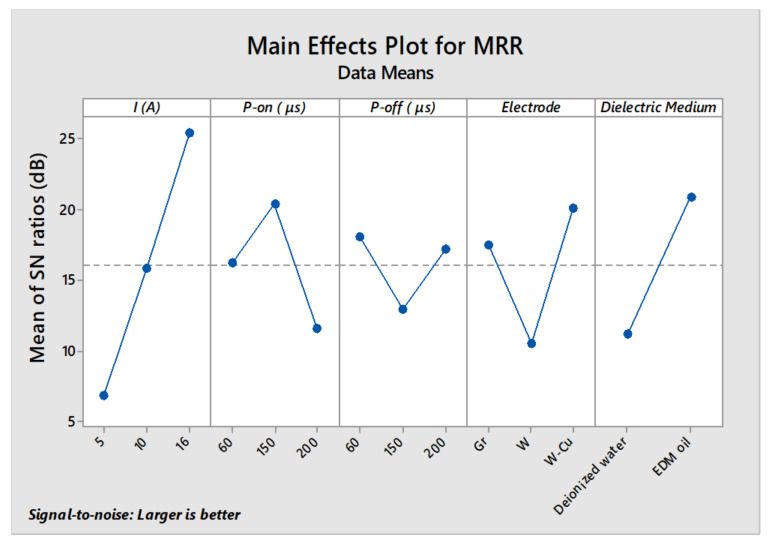
Main effect plot for S/N ratios of Material Removal Rate.

**Figure 3 micromachines-11-00926-f003:**
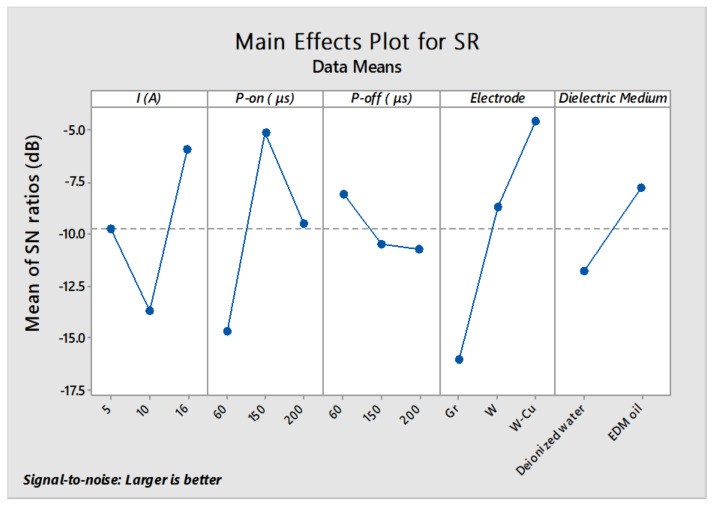
Main effect plot for S/N ratios of Surface Roughness.

**Figure 4 micromachines-11-00926-f004:**
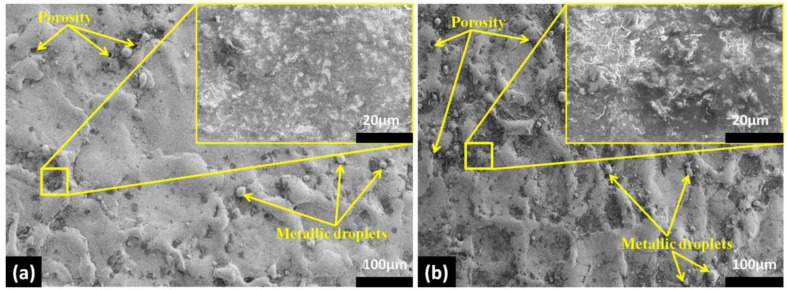
FE-SEM images illustrate surface roughness (**a**) Sample 8 (Ra = 1.4 µs), and (**b**) Sample 9 (Ra = 1.21 µs).

**Figure 5 micromachines-11-00926-f005:**
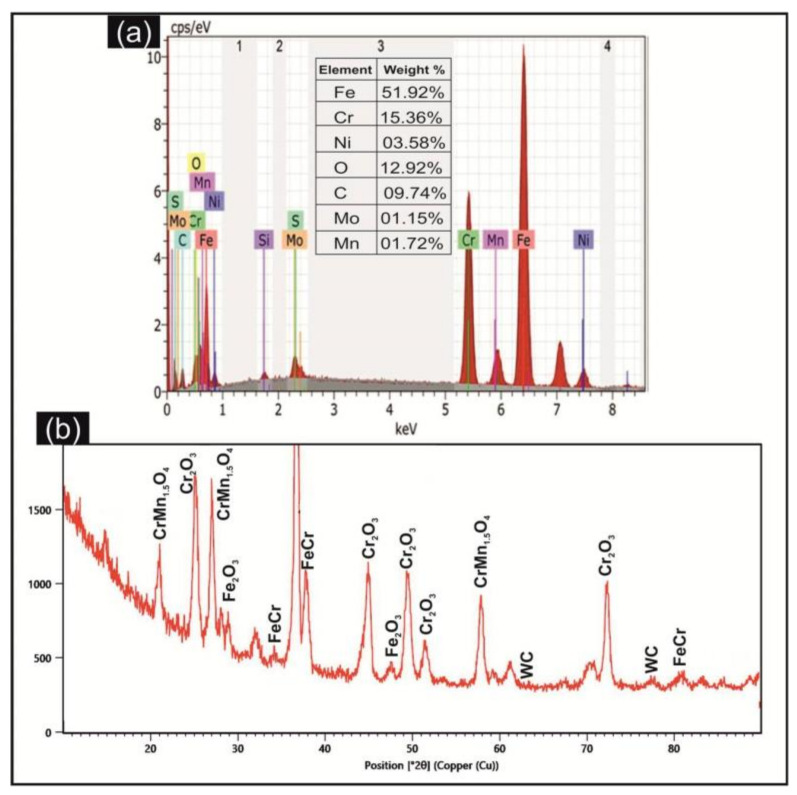
(**a**) EDS elemental spectrum and (**b**) XRD pattern of EDMed substrate.

**Figure 6 micromachines-11-00926-f006:**
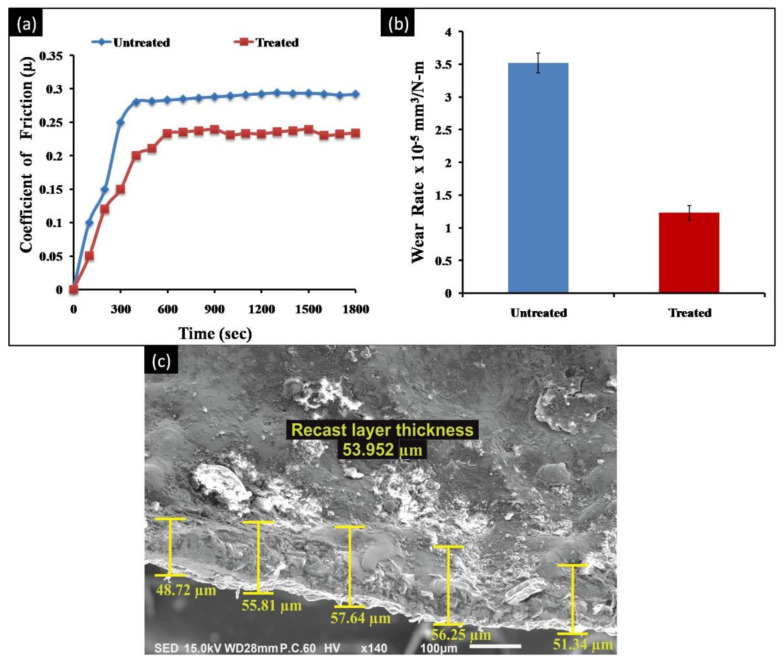
(**a**) Wear rate comparison of untreated and treated samples; (**b**) Variation of the co-efficient of friction of samples; (**c**) Cross-section of the EDMed substrate.

**Figure 7 micromachines-11-00926-f007:**
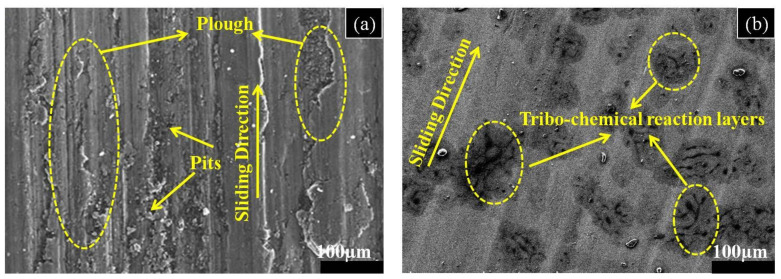
FE-SEM images represent the wear appearances of (**a**) Untreated substrate; (**b**) EDMed substrate.

**Figure 8 micromachines-11-00926-f008:**
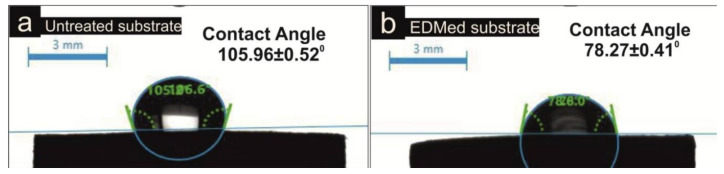
Contact angle illustration of (**a**) Untreated surface; (**b**) EDMed surface.

**Table 1 micromachines-11-00926-t001:** Properties of Graphite, Copper-Tungsten and Tungsten electrodes.

Property	Graphite	Copper-Tungsten	Tungsten
Diameter (mm)	10	10	10
Density (g/cm^3^)	2.26	14.5	18.8
Melting point (°C)	3650	3410	3400
Electrical resistivity (Ω cm)	6.0 × 10^−3^	4.5	5.6 × 10^−3^
Thermal conductivity (W/mK)	24	189	163.3
Thermal expansion coefficient (µm/mK)	6	11.7	4.5
Specific heat capacity (J/Kg °C)	720	214	133
Hardness (HB)	10	195	2570

**Table 2 micromachines-11-00926-t002:** Parameters descriptions and values.

Parameter	Units	Levels		
		Level 1	Level 2	Level 3
Current (I)	ampere	5	10	16
Pulse-on-time (P-on)	µ-seconds	60	150	200
Pulse-off-time (P-off)	µ-seconds	60	150	200
Electrode	–	Gr	W	W-Cu
Dielectric medium	–	EDM oil	Deionized water	–

**Table 3 micromachines-11-00926-t003:** Experimental L18 array design matrix and output response observations.

Exp. Trial	Levels of Controllable Parameters					Output Responses			
	I (A)	P-on (µs)	P-off (µs)	Electrode	Dielectric Medium	MRR (mm^3^/min) Avg. ± SD	S/N Ratio (MRR)	SR (µm) Avg. ± SD	S/N Ratio (SR)
1.	1	1	1	1	1	4.06 ± 0.23	12.1585	0.14 ± 0.05	−17.68
2.	1	2	2	2	1	3.85 ± 0.29	11.6704	0.56 ± 0.07	−5.17
3.	1	3	3	3	1	3.98 ± 0.14	11.9995	0.71 ± 0.07	−3.00
4.	2	1	1	2	1	13.41 ± 0.96	22.5151	0.26 ± 0.02	−11.75
5.	2	2	2	3	1	18.38 ± 0.49	25.2822	1.0 ± 0.09	-0.36
6.	2	3	3	1	1	10.31 ± 0.13	20.2695	0.2 ± 0.04	−17.25
7.	3	1	2	1	1	14.49 ± 0.21	23.2229	0.13 ± 0.04	−19.02
8.	3	2	3	2	1	27.82 ± 0.04	28.8871	1.4 ± 0.08	2.79
9.	3	3	1	3	1	39.4 ± 0.98	31.9058	1.21 ± 0.06	1.65
10.	1	1	3	3	2	3.1 ± 0.05	9.8251	0.23 ± 0.04	−13.14
11.	1	2	1	1	2	4.36 ± 0.67	12.6451	0.40 ± 0.01	−7.82
12.	1	3	2	2	2	0.14 ± 0.04	−17.2078	0.30 ± 0.06	−11.44
13.	2	1	2	3	2	4.29 ± 0.6	12.5315	0.19 ± 0.05	−14.93
14.	2	2	3	1	2	5.38 ± 0.37	14.5921	0.08 ± 0.02	−22.50
15.	2	3	1	2	2	1.01 ± 0.04	0.1421	0.18 ± 0.03	−15.14
16.	3	1	3	2	2	7.45 ± 0.45	17.4242	0.28 ± 0.05	−11.27
17.	3	2	1	3	2	28.18 ± 0.66	28.9952	1.3 ± 0.08	2.24
18.	3	3	2	1	2	12.84 ± 0.30	22.1710	0.26 ± 0.02	−11.78

**Table 4 micromachines-11-00926-t004:** ANOVA for Material Removal Rate.

Source	DF	Sum of Squares	Mean Squares	F-Value	*p*-Value	% Contribution
Current	2	1036.56	518.28	18.34	0.001 ^*^	45.18
Pulse-on-time	2	232.70	116.35	4.12	0.059	10.14
Pulse-off-time	2	89.57	44.78	1.58	0.263	3.90
Electrode	2	290.77	145.39	5.15	0.037 ^*^	12.67
Dielectric medium	1	418.50	418.50	14.81	0.005 ^*^	18.24
Error	8	226.05	28.26			
Total	17	2294.14				

^*^ Significant at 95% confidence level

**Table 5 micromachines-11-00926-t005:** ANOVA for Surface Roughness.

Source	DF	Sum of Squares	Mean Squares	F-Value	*p*-Value	% Contribution
Current	2	180.62	90.31	8.31	0.011 ^*^	17.39
Pulse-on-time	2	271.23	135.62	12.48	0.003 ^*^	26.11
Pulse-off-time	2	25.38	12.69	1.17	0.359	2.50
Electrode	2	402.08	201.04	18.50	0.001 ^*^	38.73
Dielectric medium	1	72.03	72.03	6.63	0.033 ^*^	6.94
Error	8	86.94	10.87			
Total	17	1038.28				

^*^ Significant at 95% confidence level

**Table 6 micromachines-11-00926-t006:** Wear rate and co-efficient of friction (COF) of samples.

Sr. No.	Sample	N = 1		N = 2		N = 3		Avg. ± SD	
		COF	Wear Rate (mm^3^/Nm)	COF	Wear Rate (mm^3^/Nm)	COF	Wear Rate (mm^3^/Nm)	COF	Wear Rate (mm^3^/Nm)
1.	Untreated	0.317	3.52 × 10^−5^	0.322	3.68 × 10^−5^	0.312	3.37 × 10^−5^	0.317 ± 0.005	3.52 × 10^−5^ ± 0.15
2.	Treated	0.237	1.05 × 10^−5^	0.246	1.13 × 10^−5^	0.231	0.97 × 10^−5^	0.238 ± 0.007	1.05 × 10^−5^ ± 0.08

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
