# Peer review of "Surface Characterization and Tribological Performance Analysis of Electric Discharge Machined Duplex Stainless Steel"

_micromachines, 2020, doi:10.3390/mi11100926_

Round 1

Reviewer 1 Report

This work investigates the process parameters that affect the process performances and the surface roughness for the EDM process applied to Duplex Stainless Steel with three variant of electrode material. The manuscript is well written investigating analyzing a relatively new material for ED-machining. some aspects need to be implemented before accepting the research.

  1. I suggest to add references in two part of the introduction.

To improve the section about the determination of optimal parameters

  • Analysis of the surface quality of steel and ceramic materials machined by micro-EDM, European Society for Precision Engineering and Nanotechnology, Conference Proceedings - 18th International Conference and Exhibition, EUSPEN 2018, D’Urso G. et al.

To support the application of EDM process to several type of difficult to cut materials:

  • Machinability and energy efficiency in micro-EDM milling of zirconium boride reinforced with silicon carbide fibers, Materials, DOI: 3390/ma122333920
  • Micro-EDM milling of zirconium carbide ceramics, Precision Engineering, DOI: 1016/j.precisioneng.2020.06.002
  1. Specify all the acronyms the first time you introduce (e.g FESEM).
  2. Page 4. “Gms” is not the international system symbol used for grams.
  3. Section 2.1: Today, the EDM is not so emerging.
  4. Section 2.3: Which is the selected geometry?
  5. Section 3: How many runs were performed for each experiment trial?
  6. Section 3.1:

Why do you define two different level of significance (table 4 note)? Which is the sense of this choice?

You wrote: The S/N ratio results of MRR represented peak current as the most significant factor that majorly contributes (p-value:0.001) to remove the material from DSS-2205. It was followed by the dielectric medium (p-value:0.005), electrode material (p-value:0.037) and pulse on-time (p-value: 0.059). the effect of the pulse-off-time was negligible; consequently, it was considered as an in-significant factor for machining of DSS-2205.”

From the first part, it seems like the pulse on time affects the MRR, but it is not true (p-value>0.05). in the second part the concept is the opposite, saying that this effect is negligible. I suggest to fix this sentence.

  1. Figure 2: I suggest to divide 8(a) from (b) moving the second one to the next paragraph.
  2. Why don’t you consider for the ANOVA also the interactions? You analysis seems to be incomplete.

Author Response

List of Changes made and Answers to Reviewer’s comments

Paper Reference: micromachines-957897

Title: Surface characterization and tribological performance analysis of electric discharge machined duplex stainless steel

Authors: Timur Rizovich Ablyaz *, Evgeny Sergeevich Shlykov,
Karim Ravilevich Muratov, Amit Mahajan, Gurpreet Singh, Sandeep Devgan, Sarabjeet Singh Sidhu

Referee: 1

This work investigates the process parameters that affect the process performances and the surface roughness for the EDM process applied to Duplex Stainless Steel with three variant of electrode material. The manuscript is well written investigating analyzing a relatively new material for ED-machining. some aspects need to be implemented before accepting the research.

Authors are thankful to the reviewer for their valuable observation about the study.

Observation 1

I suggest to add references in two part of the introduction.

To improve the section about the determination of optimal parameters: -

  • Analysis of the surface quality of steel and ceramic materials machined by micro-EDM, European Society for Precision Engineering and Nanotechnology, Conference Proceedings - 18th International Conference and Exhibition, EUSPEN 2018, D’Urso G. et al.

To support the application of EDM process to several type of difficult to cut materials: -

  • Machinability and energy efficiency in micro-EDM milling of zirconium boride reinforced with silicon carbide fibers, Materials, DOI: 3390/ma122333920
  • Micro-EDM milling of zirconium carbide ceramics, Precision Engineering, DOI: 1016/j.precisioneng.2020.06.002

Suggested references have been added in the revised manuscript.

Observation 2

Specify all the acronyms the first time you introduce (e.g. FESEM).

All acronyms are demonstrated in extended format.

Observation 3

Page 4. “Gms” is not the international system symbol used for grams.

The author regrets the mistakes. “Gms” is replaced by “g”.

Observation 4

Section 2.1: Today, the EDM is not so emerging.

Amongst the non-conventional methods, Electrical discharge machining (EDM) is an efficient technique that can be utilized for machining any complex material and for producing messy geometrical shape.  Recently, the researchers highlighted that the surface alteration of bioimplants by EDM, as the futuristic method. EDM is also a proficient machining technique that has found application in the manufacturing of aerospace products, molds and dies.  These achievements reveal that the EDM is an emerging technique.

Observation 5

Section 2.3: Which is the selected geometry?

In this research, duplex stainless steel (DSS-2205) in the form of a square plate of 90 mm with a thickness of 20 mm was utilized (already mentioned in section2.1). The workpiece samples were machined with three different cylindrical tool electrodes viz. graphite (C), tungsten-copper (75-W/25-Cu) and tungsten (W) having 10mm diameter and 70mm height in die-sinking EDM. The specifications of all the three electrodes illustrated in Table 1. A constant depth of 0.5 mm was kept for all the experimental runs.

All the related information of workpiece and tool is mentioned in section 2.1.

Observation 6

Section 3: How many runs were performed for each experiment trial?

The discussions about the “repeatability of the developed samples” are included in the results and discussion section to increase the understanding of study.

Description: -

All the experimental trials were carried out thrice (18 × 3 = 54 runs) to minimize the error and for more precise outcomes. The respective results in Table 3 signified the average MRR and SR values attained from each experimental trial, followed by the standard deviation (i.e. Avg ± S. D).

Observation 7

Section 3.1: Why do you define two different level of significance (table 4 note)? Which is the sense of this choice?

You wrote: The S/N ratio results of MRR represented peak current as the most significant factor that majorly contributes (p-value:0.001) to remove the material from DSS-2205. It was followed by the dielectric medium (p-value:0.005), electrode material (p-value:0.037) and pulse on-time (p-value: 0.059). the effect of the pulse-off-time was negligible; consequently, it was considered as an in-significant factor for machining of DSS-2205.”

From the first part, it seems like the pulse on time affects the MRR, but it is not true (p-value>0.05). in the second part the concept is the opposite, saying that this effect is negligible. I suggest to fix this sentence.

The input parameters are statistically analysed and presented in Table 4. The reviewer correctly points out the significant confident level as >95%. However, in this study, the trend of input parameters on the response is emphasized instead of accuracy. Also, it noted that the EDM process is the complex machining process and it is impossible to present the linear relation of the input parameter with the response parameters. Thus, in order to develop the mathematical model, the spark energy (i.e., Spark Energy= Pon. I. Voltage) combined with the electrode materials must be considered to predict the responses. Thus, the factor with a confidence level > 90% is selected as a significant factor.

The reviewer is correct that pulse-on-time is not significant (p-value>0.05), but in second part author(s) are concluding that the effect of pulse-off-time is negligible. Both the sentences are different; however, the concern lines are re-written as per the reviewer’s comment for more clarity. Also, for better readability, single level of significance (p-value<0.05) is considered.

Observation 8

Figure 2: I suggest to divide (a) from (b) moving the second one to the next paragraph.

Figure 2 (b) is separated from Figure 2 (a). Figure 2(b) is now Figure 3.

Observation 9

Why don’t you consider for the ANOVA also the interactions? Your analysis seems to be incomplete.

The reviewer has raised a very good point in this comment. In the analysis of variance, we basically evaluate the significance of each input parameter for a particular output response, on the basis of its p-value (<0.05) and their percentage contribution. We have not reported the interaction here on the grounds of their p-values (>0.05) and correspondingly the low contribution. For the ANOVA of material removal rate (Table 4) and surface roughness (Table 5), the contribution sum of all input parameters was above 90%. There was no significant interaction as per the required p-value (<0.05), hence not reported. 

Reviewer 2 Report

The authors present a detailed study of the impacts of EDM input parameters on the tribological performance of duplex stainless steels. The ANOVA analysis was well done but the surface analysis, wettability, and tribological tests were quite sparse in terms of comparing the inputs. It would be nice to see some discussion and conclusions about how the input parameters correlate and/or cause favorable tribological features.

Introduction

The literature review does not add too much value to the paper (paragraph two specifically). It would be better to review what is known about the relationship between EDM inputs on the outputs examined in this study for comparable metals.

Results

Can you compare surface analysis, wettability, and tribological performance between trials? I think this would dramatically improve the value of this study. This could provide insights into how these inputs work together, or against each other to drive the observed tribological outcomes.

Author Response

List of Changes made and Answers to Reviewer’s comments

Paper Reference: micromachines-957897

Title: Surface characterization and tribological performance analysis of electric discharge machined duplex stainless steel

Authors: Timur Rizovich Ablyaz *, Evgeny Sergeevich Shlykov,
Karim Ravilevich Muratov, Amit Mahajan, Gurpreet Singh, Sandeep Devgan, Sarabjeet Singh Sidhu

The authors present a detailed study of the impacts of EDM input parameters on the tribological performance of duplex stainless steels. The ANOVA analysis was well done but the surface analysis, wettability, and tribological tests were quite sparse in terms of comparing the inputs. It would be nice to see some discussion and conclusions about how the input parameters correlate and/or cause favorable tribological features.

Thank you for giving us the opportunity to submit a revised draft of our manuscript. The authors are grateful to the reviewer for the insightful comments on this manuscript. We have been able to incorporate changes to reflect most of the suggestions provided by the reviewer.

Observation 1

Introduction

The literature review does not add too much value to the paper (paragraph two specifically). It would be better to review what is known about the relationship between EDM inputs on the outputs examined in this study for comparable metals.

The concern section is revised and re-written to justify the reviewer’s comment. New references are added and also the previous ones are arranged systemically.

Round 2

Reviewer 1 Report

In my opinion the authors answered satisfactorily to the comments improving what was requested. 

Only one more observation, in the paper you identify Pulse-on-time and Pulse-off--time as Pon and Poff, but generally they are identified as ton and toff.

I suggest to accept this work. 

Author Response

Thank you very much. We appreciate the time and effort that you have dedicated to providing your positive feedback on the manuscript. Response to the query about P-on and P-off, both the terms can be used to symbolize the pulse on/off duration. Many author(s) are using the same words to represent in the recently reporting articles. The reviewer is right that ton and toff are generally used, but we can also use identically the words P-on, P-off or Pulse-on, Pulse-off.

Some of the articles with similar acronyms are: -

  • Bains, P.S.; Sidhu, S.S.; Payal, H.S.; Kaur, S.  Magnetic field influence on surface modifications in powder mixed EDM. Silicon 2019,11, 415–423. doi: 10.1007/s12633-018-9907-z
  • Singh, G.; Sidhu, S.S.; Bains, P.S.; Bhui, A.S. Improving microhardness and wear resistance of 316L by TiO2 powder mixed electro-discharge treatment. Res. Express. 2019, 6, 086501, doi:10.1088/2053-1591/ab1bab
  • Niamat, M.; Sarfraz, S.; Aziz, H.; Jahanzaib, M.; Shehab, E.; Ahmad, W.; Hussain, S. Effect of different dielectrics on material removal rate, electrode wear rate and microstructures in EDM. Procedia CIRP 2014, 60, 2–7. doi: 10.1016/j.procir.2017.02.023 

Reviewer 2 Report

Surface analysis results and analysis are still light. No clear connection between their test variables and their impact of surface features.

Author Response

The surface morphology section is revised and updated to improve the manuscript as per the reviewer’s comment. The connection of input process parameters affecting the surface is discussed and highlighted in the revised manuscript.

Description: -

Both the machined substrates showed micro- macropores and re-solidified metallic droplets on the surface. It is observable from the images (Figure 4a and 4b) that higher spark energy (Spark Energy = Current × P-on × Voltage) generate micro-macropores along with small peaks and valleys on the EDT surface [1,2].

  1. Cogun, C.; Esen, Z.; Genc, A.; Cogun, F.; Akturk, N. Effect of powder metallurgy Cu-B4C electrodes on workpiece surface characteristics and machining performance of electric discharge machining. IMechE Part B: J. Eng. Manuf. 2016, 230, 2190–2203. doi: 10.1177/0954405415593049 
  2. Ji, R.; Liu, Y.; Diao, R.; Xu, C.; Li, X.; Cai, B.; Zhang, Y. Influence of electrical resistivity and machining parameters on electrical discharge machining performance of engineering ceramics. PLoS ONE 2014, 9, e110775. doi: 10.1371/journal.pone.0110775 

Authors sincerely thank the reviewer(s) for their valuable comments and suggestions that helped to

improve the quality of the paper.

Thanks & Regards